# Analysis and Visualization of Research on Resilient Cities and Communities Based on VOSviewer

**DOI:** 10.3390/ijerph19127068

**Published:** 2022-06-09

**Authors:** Yu-Jie Huang, Shuo Cheng, Fu-Qiang Yang, Chao Chen

**Affiliations:** 1College of Environment and Safety Engineering, Fuzhou University, Fuzhou 350116, China; huangyujie199871@163.com (Y.-J.H.); jessecheng712@gmail.com (S.C.); 2Safety and Security Science Group, Faculty of Technology, Policy and Management, TU Delft, 2628 BX Delft, The Netherlands; 3School of Petroleum Engineering, Southwest Petroleum University, Chengdu 610500, China

**Keywords:** resilient cities, resilient communities, bibliometrics, knowledge graph, VOSviewer

## Abstract

To objectively grasp the current situation and development trend of resilient cities or communities (RC) research. The articles in Web of Science (WOS) Core Collection databases from 1995 to 2022 were used as a sample, and bibliometrics was used to statistically analyze the year of publication and number of articles, highly cited documents and keyword hotness in this field. VOSviewer was used to explore the knowledge graph of RC research documents. The results show that: the development process is roughly divided into 3 periods: no attention (1995–2004), starting (2005–2014), and rapid growth (2015–2021). The journal “*Sustainability*” and “*International journal of disaster risk reduction*” are the key journals publishing RC research. Serre and Shaw are the most productive authors. The USA is still the leading country in this field of RC. Colorado State Univ, Texas a&m Univ, and Delft Univ Technol are the main research institutions. The keyword analysis indicates the hot topics in different periods. Moreover, several limitations and some recommendations for future research on RC are also given based on this.

## 1. Introduction

Firstly, the concept of “resilience” appeared in ecology in the 1970s [1] and has evolved from “stability” to “adaptability”. Initially, resilience was defined as the ability to cope with changes and resist disturbances while maintaining the same basic state [2]. Its conceptual development went through stages of development from ecological resilience, engineering resilience, and then evolutionary resilience [3,4,5,6]. Ecological resilience emphasizes the ability to adapt to external shocks and control interactive change [7,8]. Engineering resilience concentrates on the stability of the physical system, emphasizing its ability to return to its pre-disturbance state [9]. In the social economy, psychological resilience [10], organizational resilience [11], and industrial resilience [12] were proposed from the perspective of the system level. Nowadays, it is mainly applied in industries such as disaster and climate issues, the resilience of key cities and regional economic development, the resilience of key urban infrastructure development, the resilience of large cities to terrorist events, indoor spaces, and urban planning and construction. The concept of resilience, with its connotations of dynamism, co-evolution, and “bouncing to a better state” [13,14], has been widely applied to the study of adaptation strategies of urban systems in the face of large and uncertain changes that cannot be fully predicted in the future [15]. It has become the focus of current research [16].

Today, rising global temperatures, extreme climate disasters, and urban terrorism are constantly threatening cities. For example, the stampede on the Bund in Shanghai in December 2004, the extraordinarily heavy rainstorm in Beijing in July 2012, the mega-storm Sandy on the USA West Coast in November 2012, the Deadly Forest Fire in California USA in November 2018, and the COVID-19 outbreak in 2020, which caused a large number of casualties and huge economic losses and had a very bad social impact. The outbreak of major public health emergencies in early 2020 (COVID-19) has resulted in more than 480.17 million confirmed cases and 6.12 million deaths worldwide until 29 March 2022. With the uncertainty of exogenous risk perturbations and the continuous variability of endogenous urban structures, inadequate emergency preparedness and response capabilities often result in huge losses, and there is an urgent need to improve urban resilience to cope with various risk perturbations.

RC has a become practical guideline for preventing and resisting disaster risks in cities and communities. It has long been a common concern of scholars worldwide. In this social context, it is of great academic and social value to establish a framework for RC research, explore methods for enhancing urban resilience, and improve evaluation criteria for RC. Therefore, it is necessary to sort out their overall research characteristics and trends. Research on RC has been conducted at home and abroad, most of which focus on a certain aspect, such as the evolutionary mechanism of resilient cities [17], the evolution of concepts and theories [18,19], spatial resilience [20], and quantitative assessment of resilience [21,22], etc. Most of the existing studies are qualitative and relatively small in span, and there is a lack of comparative analysis of domestic and international studies.

One important part of bibliometrics is citation visualization analysis methods which has gradually developed in the context of scientometrics and data visualization, revealing the intrinsic connections and research regulations of disciplines in the form of knowledge mapping [23]. As a quantitative analysis tool, the bibliometric approach is applied to understand the current status and gaps in a certain field [24,25,26,27]. Currently, the bibliometric visualization analysis method has been employed in a wide variety of research fields. For example, Su et al. [28] conducted a visualized bibliometric analysis to map the research trends of machine learning and burst detection analysis to show the result in engineering (MLE) based on articles indexed in the WoS Core Collection published between 2000 and 2019. Paulo et al. [29] applied bibliometric analysis to evaluate the global scientific production on ecological restoration from the period from 1997 to 2017. Anugerah et al. [30] analyzed Social Network Analysis (SNA) approach in business and management research from the Scopus database published from 2001 to 2020. Wu et al. [31] combined bibliometric analysis and network analysis to explore urban community governance research.

Yang et al. (2021) [32] applied Citespace and VOSviewer to analyze the research progress of resilient cities published in the Web of Science database from 2010 to 2019. Yang took a different approach, in particular restricting their time span to publications in the years after 2010, as there were too few papers from the previous year. However, it was necessary to start the study with the first paper in the field. Moreover, they used a different retrieval method by keywords (Resilience city/resilience urban/Resilient city/resilient urban). This is helpful for scholars to explore the current research progress in the field of resilient cities, but it also has some shortcomings. For example, the search formulation fails to take into account the plural form of “city” or “urban”. Compared with the previous work, this paper selected a longer time span (from 1995 to 2022) and took into account the plural form of the retrieved keywords. This is more conducive to a comprehensive understanding of the evolutionary trends in the field of resilient cities.

In this paper, the bibliometric method is applied in the area of RC, and insights in this research field are obtained based on publication records retrieved from the WoS Core Collection. The objective of this study is to provide a macroscopic overview of the main characteristics of RC publications and a clear picture of the research process in the field of RC research. The total profile of yearly output, cooperation networks, citation performance, research hotspots, and development trends are recognized. The remainder of this paper is organized as follows. Section 2 presents the data sources, the software of analysis, and the bibliometric method. Section 3 introduced the results of the analysis. Section 4 presents the conclusions and discussion. 

## 2. Materials and Methods

The data used in this research are obtained from the Web of Science Core Collection database. The search work was finished on 16 April 2022. WoS database is a scientific publishing research database that is widely used by researchers all over the world [33]. The search rule: TS = (“resilient cit*” or “resilient communit*”). The timespan was set from 1995 to 2022. We excluded some irrelevant literature such as book indices. Finally, a total of 1148 documents were obtained. The documents were exported as “plain text formats”.

Bibliometrics is a quantitative analysis method of paper using mathematics, applied statistics, and other research ideas. VOSviewer is a bibliometric software, which introduces sample pieces of literature data into it and draws knowledge maps. It can present the overall external characteristics of subject areas, and the software has unique advantages, especially in clustering analysis [34]. This paper applies VOSviewer to conduct a bibliometric analysis of the research papers on resilient cities or resilient communities at home and abroad from 1995 to 2022 in order to explore their research hotspots and development trends. VOSviewer runs in Java environment [35] and can import pieces of literature in Web of Science formats. The used research procedure and method are shown in Figure 1.

## 3. Results 

### 3.1. The Growth and Output Publications

The year of publication and the number of publications reflects the overall trend, development speed, and research hotspots in this field to a certain extent [36]. The number of publications related to RC retrieved from WoS was counted. As shown in Table 1, 11 types of documents exist.

The yearly number of publications and cumulative number of RC from 1995 to 2022 are shown in Figure 2. Overall, the number of articles on RC dramatically increased from the earliest in 1995 to 213 articles by 2021. It can be illustrated that the significance and attention of RC research have increased. From 1995 to 2004, there were almost no relevant articles published in this field. After 2004, the number of research papers in this field showed a general trend of steady growth. Its development process can be roughly divided into 3 stages. In the period 1995 to 2004, the number of published articles was low, which means that any attention had not been received to resilient cities or resilient communities. The first period is classified as starting stage. In the period from 2004 to 2014, the number of the published article presents a trend of stable increase, with exceptions in 2008, 2012, and 2014, where a decrease can be found. Between 2014 and 2021 represents the third stage. With the acceleration of global urbanization and the increase in various natural and human disasters, the vulnerability of cities has become increasingly obvious. The third period belongs to the rapid growth stage. The scholars carried out more exploration and research according to the uniqueness of urban culture, and the number of articles increased to 213 in 2021.

### 3.2. Source Distribution of Publications

The analysis of publication sources is a valuable method for distinguishing the core journals related to RC, and it is of great importance for scholars to search for related pieces of literature and choose an appropriate journal. According to the retrieved results, a total of 627 journals published RC research from 1995 to 2022. Table 2 provides information about the rank of active journals. It can be seen that *Sustainability* is the most productive journal, followed by the *International Journal of disaster risk reduction*, *Cities*, *Sustainable cities and society*, and *natural hazards*. The journal impact factor is a measure of a journal’s impact capacity. As to the impact factor in 2020, *the Journal of cleaner production* is the most influential journal in RC research, followed by *Sustainable cities and society*. The results also indicate that RC is a multidisciplinary research field with a multitude of disciplines such as environmental science, disaster risk prevention, urban planning, and sustainable development.

### 3.3. Collaboration Networks

According to the analysis of the database and statistics on the number and distribution trends of the authors in core journals, the innovative and highly productive talents in the field and the cooperation status between their research teams can be judged [23]. The core researchers in the field and the influence of the personnel can be quickly understood. Based on the statistics of the authors in the field of RC in the WoS database, 1148 papers involving 3464 authors can be retrieved, and the VOSviewer was used to cluster the authors and draw the cooperation network of authors with more than 2 documents. As shown in Figure 3, the circles represent authors, the size of circles is positively correlated with the connection strength of authors, authors with the same color in the view belong to the same clustered cooperation network, and the line means the connection strength between different authors. In the cooperation network, eleven major clusters of authors can be distinguished. Among them, the red cluster has the most collaborators including Shaw, van de, Wilkinson, Khatibi, Hernantes, and Labaka. Followed by the blue cluster and the green cluster. The main researchers in the network are Wilkinson, Dianat, Khatibi, and Meerow. Other researchers are linked to one of these main researchers.

To determine the most frequently appearing authors in RC research, we analyze the country and institute of authors, number of documents published, average citations per publication, and H-index. Table 3 shows the top 10 authors that published the most articles on RC. Serre and Shaw are the authors with the largest number of documents on RC. Interestingly, 3 of the top 10 authors come from the USA and 3 come from France. In terms of average citation and H-index, Stults, Berke, Diab, and Serre are the most meaningful scholars. Besides, Figure 3 indicates that many scholars still publish independently and have less contact with other teams. Teams tend to be less connected and their cooperation is more scattered. 

The collaboration network between different institutions on RC research is shown in Figure 4. Figure 4 indicates that the organizational collaboration network on RC contains 215 new items, 25 clusters, 397 links, and 434 total link strengths. In terms of the number of links, Colorado State Univ and University College London’s links are the most which both have 16 lines and the most cooperation with others in RC research; this is followed by Stockholm Univ (links = 15), Univ Calif Berkeley (links = 12), and Natl Univ Singapore (links = 12). According to the number of documents published, the institution which published the most publications on the topic are the Colorado State Univ, Texas a&m Univ, and Delft Univ Technol, which have the same documents in RC research (n = 14). According to the link strength, Colorado State Univ has the closest collaboration in RC research, followed by University College London.

It is remarkable that the cooperative networks have small group characteristics, for example, Islamic Azad Univ, Kharazmi Univ, Sharif Univ Technol, and Univ Tehran. We can see that there is not enough academic cooperation between the various institutions. Therefore, it is necessary to build a multi-center cooperation network.

Figure 5 shows the number of documents, citations, and total link strength with the top 17 institutions according to the publications. Eight of the seventeen institutions are located in the USA and one institution is from China. Univ British Columbia and Univ Waterloo are both situated in Canada. Univ Melbourne and Univ Sydney are both situated in Australia. Colorado State Univ is classified with 14 papers and 408 citations. Delft Univ Technol has the most citations (713), followed by Univ British Columbia (686). Surprisingly, these two institutions are not all from the USA. 

### 3.4. Citation and Co-Citation Analysis

The analysis of highly cited documents helps us to understand the knowledge base and the development of the field. The citation frequency of documents in the field of RC in the Web of Science Core Collection database from 1995 to 2022 was counted. The minimum citation frequency was set to 30, and a total of 175 papers complied with the set requirements, which were screened and divided into 23 clusters. Co-citation analysis focuses on the relationship or interaction between two publications and gives an overview of publications that have been cited together in other publications. The more often two publications are cited together, the greater the similarity between them can be deemed [36,37].

The most frequently cited publications in this field of RC are shown in Figure 6. Holling (1973) [1]: Resilience and Stability of Ecological Systems has the highest total link strength (516), followed by Folke (2006) [38]: Resilience: The emergence of a perspective for social-ecological systems analyses, and Godschalk (2003) [39]: Urban hazard mitigation: creating resilient cities. The total link strength refers to the total number of co-occurrences of the node with other nodes (including repeated co-occurrences), however, this is influenced by the number of authors with whom it collaborates. In general, the author published Resilience and Stability of Ecological Systems is more associated with other collaborators. The most cited paper according to the frequency of documents cited is published by Adger [40] in 2000 titled “Social and ecological resilience: are they related?”, with a total of 1978 times cited. The paper from Adger (2000) can be regarded as a significant influence in this field. “Defining urban resilience: a review” (Meerow, 2016b) [41] published by Landscape and urban planning, and “Urban Hazard Mitigation: Creating Resilient Cities”, authored by Godschalk (2003) [39] in Natural hazards review are ranked No. 2 and No. 3 as most highly cited articles, with, respectively, 734 and 632 citations. It should be observed that there is a general hypothesis that the number of the citations indicates the influence, notoriety, and its quality of a publication [42,43]. However, Walter et al. [44] have noted that the times when others have cited publications do not indicate the quality of a publication, but rather measure its visibility. In addition, there is a growing awareness that open access journal publications are increasingly cited [45]. Older publications have more chance to be already cited than newer publications, but this does not preclude recent publications from having a critical influence in this domain. For example, the most recently published paper in 2016 has been cited 734 times, compared to the earliest published paper in 2003 which has been cited 632 times.

As shown in Figure 6, the size of the circles suggests the number of citations; the larger a circle, the more a document has been cited in the RC domain. A smaller distance between two publications represents a stronger connection and a higher similarity between them. Circles with the same color represent a similar theme among these papers. Figure 6 illustrates three distinct clusters, where each cluster represents a field of RC research: green cluster, blue cluster, and red cluster. The green cluster concentrates on the construction and planning of urban resilience. The blue cluster focuses on the ecological system resilience. The red cluster preferred the research on social resilience.

### 3.5. Keyword Hotness Analysis

Keywords are the authors’ high overview of research papers, and keyword analysis of research papers in a certain field can quickly locate the research hotspots and frontiers in the field [23]. A total of 1148 documents retrieved in WoS were imported into VOSviewer, a total of 83 keywords with 12 or more occurrences were selected. The keywords were analyzed by clustering, and the keyword co-occurrence mapping in the field of resilient cities or resilient communities was drawn, as shown in Figure 7.

The nodes in the figure are keywords. The size of the nodes indicates the frequency of the keywords. The connecting lines indicate the co-occurrence of the keywords. According to the different colors of keyword nodes in Figure 7, keywords can be divided into three different topics (excluding small clusters). As is shown in Figure 7 there are three distinct term clusters, which are, respectively, depicted in cluster one (green color network), cluster two (blue color network), and cluster three (red color network). 

The green cluster (cluster 1) includes 27 keywords. This cluster consists of keywords such as management, sustainability, climate-change, infrastructure, challenges, biodiversity, impact, and ecosystem services. The most frequent keyword in the green cluster is “climate-change”, followed by “management”. We can see that the green cluster spreads out around “climate change” and “management”. Accordingly, it can be inferred that these publications focus on biological ecology, natural environment, and resilient cities’ management.

The blue cluster (cluster 2) is mainly around the keyword “cities”, whose frequency is the first among all the keywords. The other main keywords in the blue cluster refer to, resilient city, urban planning, framework, urban, systems, policy, smart city, and COVID-19. Namely, the blue cluster concentrates on the planning and design of urban.

The red cluster (cluster 3) is the largest and consists of 30 keywords. The most collaborative keywords are resilience, adaptation, community resilience, vulnerability, disaster, health, governance, and recovery. This analysis reveals that this cluster concentrates on disaster prevention and risk reduction.

Figure 8 shows the analysis of the keywords of RC documents over time. The term’s average publication year is demonstrated by the color of a term. Moreover, the density view of hot terms on RC during 1995–2004, 2005–2014, and 2015–2022 are given to grasp the temporal evolution in this field (see Figure 9). Additionally, Table 4 lists the top 20 keywords in RC research from 1995 to 2022. From Table 4, we can see that few relevant articles have been published before 2004. It can be seen that most research before 2014 focused on the climate and nature of resilient cities (the hot topics including adaptation, vulnerability, management, resilient cities, climate change, disaster, and sustainability). In the most recent years, the notable research topics concentrated on the climate-change, framework, and urban resilience.

## 4. Conclusions 

In recent years, both literature and practice cases in the field of RC have been increasing and the research results and practical experiences have received wide attention. This paper uses VOSviewer to visualize and analyze the year of publication and number of articles, keyword hotness, research authors and cooperation networks, representative research institutions, and highly cited documents of RC research. The study includes 1148 publications on RC covering 3464 authors, 627 journals, and 1464 institutions. 

The analyses provided information on who is standing on the frontier of this research area:(1)The development process is roughly divided into 3 periods: no attention (1995–2004), starting (2005–2014), and rapid growth (2015–2021). After 2014, the research documents in this field showed a steady increase. Generally, a variety of RC studies have attracted an increasing concern in recent years;(2)The journal “*Sustainability*” and “*International journal of disaster risk reduction*” are the key journals which have published resilient cities or communities research;(3)The USA, Canada, and Australia are the countries dominating the publication production. Colorado State Univ, Texas a&m Univ, and Delft Univ Technol have the most documents in RC research publishing 14 papers. A total of 8 of the top-17 institutions are located in the USA. The USA is still the leading country in this field of RC. Serre and Shaw are the most productive authors. Results indicate that many scholars still publish independently and have less contact with other teams;(4)The most cited paper is from Adger (2000) titled “Social and ecological resilience: are they related?”, Holling (1973): Resilience and Stability of Ecological Systems has the highest total link strength. A quick overview of the research in this field can be obtained from literature;(5)A keyword analysis indicates that most of the research before 2014 focused on adaptation, vulnerability, management, resilient cities, climate change, disaster, and sustainability. In the most recent years, the notable research topics concentrated on the climate-change, framework, and urban resilience.

Based on the results of the above, it can be seen that most institutions were universities, lacking in-depth cooperation with enterprises and governmental organizations. In the field of future RC research, we should strengthen the cooperation between industry, academia, and governmental organization, which can stimulate innovative thinking and make the research results more relevant and practical. Additionally, this can promote the diversification of research hotspots and detailed research directions in the field of resilient cities and produce more breakthrough research results.

Finally, several limitations of this study should be addressed. Firstly, the search was limited to documents published in WoS. Although WoS is one of the largest global databases; it does not include all documents in the domain of RC research. Other international databases such as PubMed, Scopus, and Sprint could be used in combination. Additionally, some publications were incorporated, although they are involved in the subject of RC. Finally, the paper can only analyze existing categories in the Web of Science, leading to the omission of some information in empirical research, such as background details (for example, the departments). Based on the limitations of this study, we recommend that a deeper content analysis is conducted.

## Figures and Tables

**Figure 1 ijerph-19-07068-f001:**
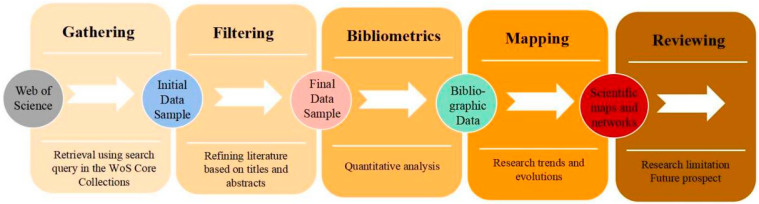
The research procedure and method.

**Figure 2 ijerph-19-07068-f002:**
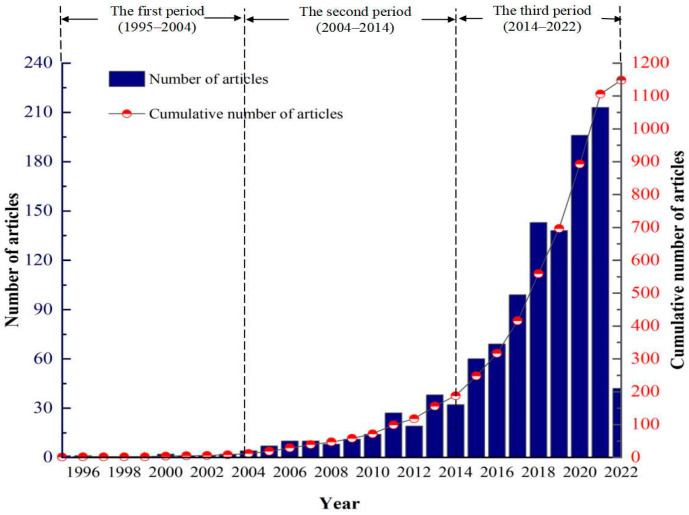
The yearly publication and number of publications on RC from 1995 to 2022.

**Figure 3 ijerph-19-07068-f003:**
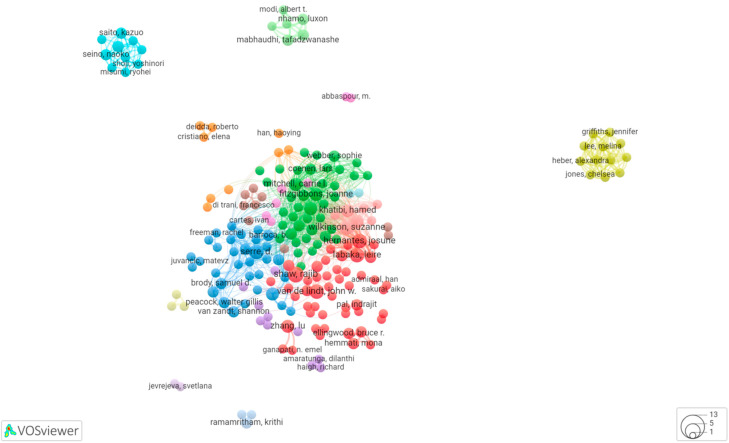
Map of the authors and collaboration network.

**Figure 4 ijerph-19-07068-f004:**
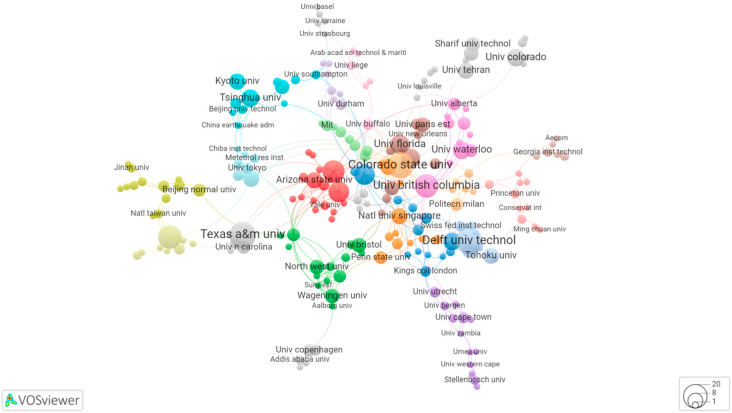
Collaboration network between different institutions on RC research.

**Figure 5 ijerph-19-07068-f005:**
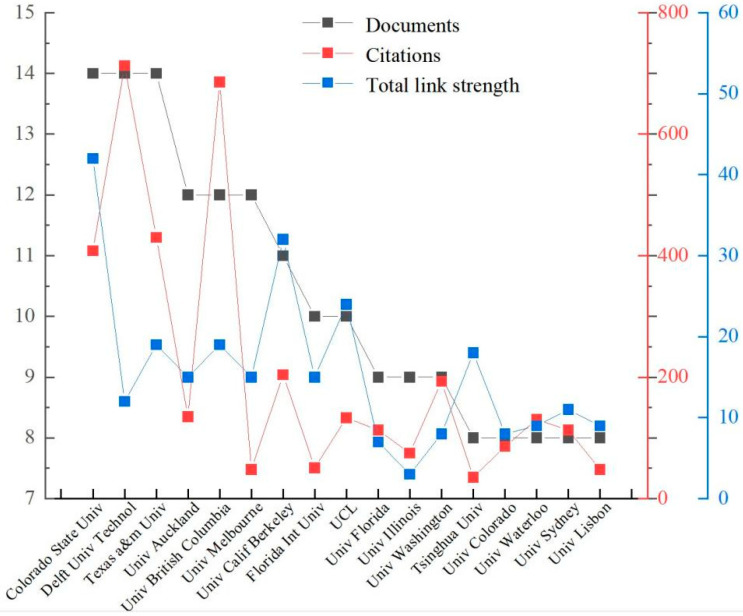
Top-17 of most productive institutions publishing on RC.

**Figure 6 ijerph-19-07068-f006:**
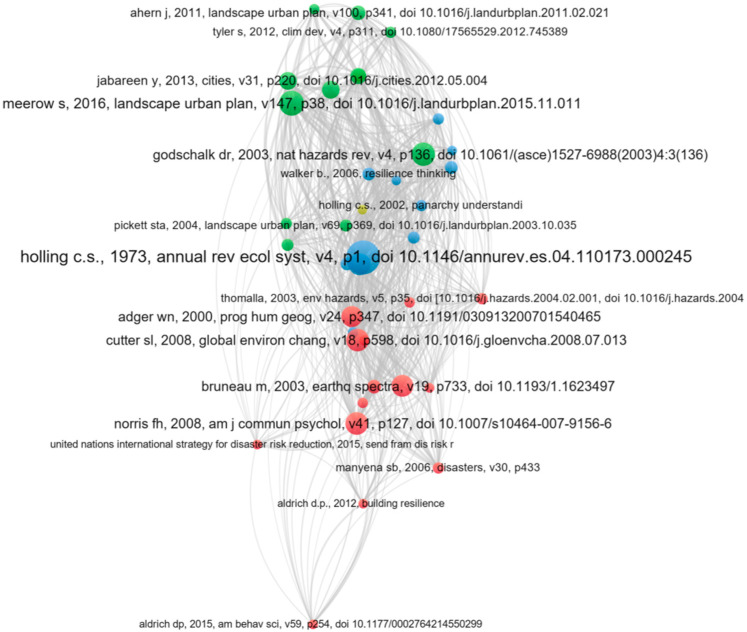
The most frequent cited publications in this field of RC.

**Figure 7 ijerph-19-07068-f007:**
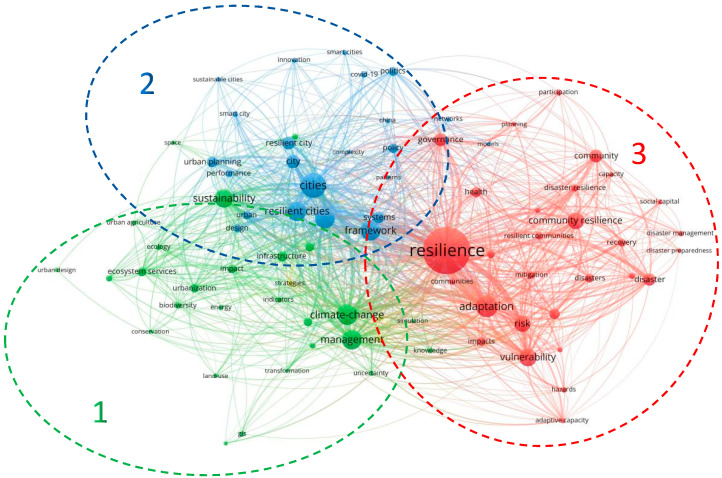
Keywords co-occurrence analysis of RC publications.

**Figure 8 ijerph-19-07068-f008:**
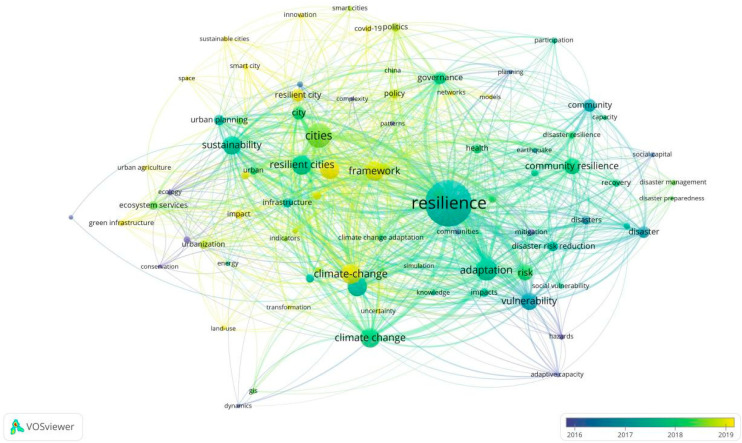
Keywords analysis of RC publications with time information.

**Figure 9 ijerph-19-07068-f009:**
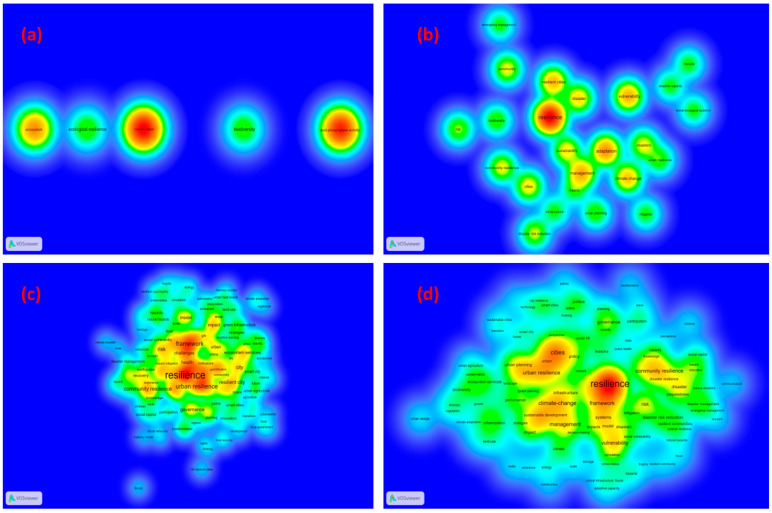
Research topic evolution over time in RC research: (**a**) 1995–2004, (**b**) 2005–2014, (**c**) 2015–2022, (**d**) 1995–2022.

**Table 1 ijerph-19-07068-t001:** The number of RC documents in each category during 1995–2022.

Rank.	Type of Documents	No. of Documents	Percentage
1	Article	841	73.2%
2	Conference record	183	15.9%
3	Review	53	4.6%
4	Book review	37	3.2%
5	Editorial material	35	3.0%
6	Publish online	30	2.6%
7	Meeting abstract	7	0.6%
8	Book chapter	4	0.3%
9	Revise	1	0.1%
10	Data paper	1	0.1%
11	News item	1	0.1%
Total	--	1148	--

**Table 2 ijerph-19-07068-t002:** The rank of active journals about publications of RC (Top 12).

Journal	Documents	Proportion (%)	Citations	Impact Factor in 2020
Sustainability	58	5.1	510	3.251
International journal of disaster risk reduction	34	3.0	448	4.320
Cities	29	2.5	1057	5.835
Sustainable cities and society	27	2.4	227	7.587
Natural hazards	19	1.7	485	3.102
7th international conference on building resilience: using scientific	13	1.1	41	-
Landscape and urban planning	12	1.0	1532	6.142
International journal of environmental research and public health	11	0.9	70	3.390
Scientia Iranica	10	0.9	39	1.435
Water	10	0.9	77	3.103
Disaster prevention and management	9	0.8	102	1.521
Journal of cleaner production	9	0.8	584	9.297

Impact factors were retrieved from the 2021 Journal Citation Reports.

**Table 3 ijerph-19-07068-t003:** The top 10 prolific authors on RC.

Author	Country/Institute	Documents	Average Citations Per Publication	H-Index
Serre, D	France/Avignon University	6	26.67	5
Shaw, R	Japan/Keio University	6	12.17	4
Berke, PR	USA/University of North Carolina	5	42.2	3
Hernantes, J	Spain/University of Navarra	5	16	4
Labaka, L	Spain/University of Navarra	5	16	4
Stults, M	USA/University of Michigan	5	163.4	4
Van De Lindt, JW	USA/Colorado State University	5	9.6	4
Wilkinson, S	New Zealand/Massey University	5	4	3
Diab, Y	France/University Gustave-Eiffel	4	27.25	3
Barroca, B	France/University Gustave-Eiffel	4	17.5	4

**Table 4 ijerph-19-07068-t004:** Distribution of top 20 keywords of RC.

Rank	1995–2004		2005–2014		2014–2022		Global	
	Terms	F	Terms	F	Terms	F	Terms	F
1	Diversity	2	Resilience	39	Resilience	190	Resilience	230
2	Biodiversity	2	Adaptation	18	Cities	96	Cities	106
3	Ecological resilience	2	Vulnerability	15	Climate-change	81	Adaptation	88
4	Brazil	1	Management	15	Framework	75	Climate-change	83
5	Cerrado	1	Resilient cities	14	Urban resilience	74	Management	80
6	Conservation	1	Climate change	13	Adaptation	70	Urban resilience	80
7	Cultural geography	1	Disaster	11	Resilient cities	66	Resilient cities	80
8	Dynamics	1	Cities	10	Management	65	Framework	79
9	Environment	1	Sustainability	10	Climate change	65	Climate change	78
10	Fire	1	Community	9	Sustainability	62	Sustainability	73
11	Floristics	1	Community resilience	9	Community resilience	55	Vulnerability	70
12	Game gallery forest	1	Disasters	9	Vulnerability	54	Community resilience	64
13	Growth	1	Disaster risk reduction	8	Risk	53	Risk	60
14	Human ecology	1	Risk	7	City	45	City	49
15	Inequality	1	Urban resilience	6	Resilient city	45	Governance	46
16	Knowledge	1	Urban planning	6	Governance	42	Community	46
17	Mortality	1	Biodiversity	6	Systems	40	Resilient city	46
18	Permanent plots	1	Infrastructure	5	Community	37	Disaster	43
19	Recruitment	1	Impacts	5	Model	35	Systems	42
20	Resource	1	Social-ecological systems	5	Health	33	Urban planning	37

## Data Availability

Not applicable.

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
