# Peer review of "Analysis and Visualization of Research on Resilient Cities and Communities Based on VOSviewer"

_ijerph, 2022, doi:10.3390/ijerph19127068_

Round 1
Reviewer 1 Report
The current revision is a very different manuscript, has been improved in some aspects, but still has significant deficits in methodology and presentation.
The main differences/improvements/criticism:
- The authors dropped their focus on Chinese research retrieved from the Chinese database CNKI which makes the investigations more reproducible. The title is changed accordingly. But it is still not as clear as it should be. I think the authors mean this: “Analysis and visualization of research on resilient cities and communities based on VOSviewer”.
- 1. Introduction: This part is significantly improved and now contains many more helpful explanations and references on the topic of resilience as well as on the bibliometric methods that the author are going to use and for what purpose. But on line 64 I suggest a different wording for clarity: “One important part of bibliometrics are citation visualization analysis methods gradually …”
And one important reference is missing: The review article “Resilient City: A Bibliometric Analysis and Visualization” by Yang et al (2021), https://doi.org/10.1155/2021/5558497. Yang et al. seem to have undertaken very much the same endeavour as the present authors and it is a “must” to interact with their approach, their methods (retrieval formulation, visualization using CiteSpace) and obviously different results(!). The authors must give clear reason why they even performed their investigations in the first place, when there already was such a review, and what they wanted to improve or which was their different scope. I’m sure the authors knew this work, because they nearly cite it on lines 127 to 129 and have the same division in time periods from 2010 on in Fig 3 and on lines 125 to 127. But they are not giving the reference which is no good scientific practice!
- 2. Materials and Methods: For the search procedure in WoS, the authors took my suggestion and described it in a bit more detail. But a screenshot of the WoS web interface has no additional information content! And my suggestion was by no means meant as a comprehensive search strategy in order to maximize recall and precision of the topical search. This method of retrieval is still a main weakness. Again my own retrieval on 19 May 2022 (restricted to indexing dates until four weeks ago) got different results: more than 1,300 documents. So the authors applied some restriction, probably excluding the book indices, which they should mention and explain. Then Table 1, listing the different document types, is comparable to my results. Moreover, it is better practice to restrict the publication year to the last full year for bibliometric studies, in this case 2021. This makes the results more reproducible and presentable, cf. Fig 3.
Their research procedure is now clearer described in Figure 2 (in the first box I would not talk of “data mining”, it is simple “retrieval”).
As mentioned above Yang et al (2021) have a different approach, in particular restricting their time span to publications years after 2010, because the former year have too few papers. Moreover, they use a different retrieval method using some keywords. Trying to reproduce their search and also excluding the book indices I got more than 1,500 papers between 2010 and 2019– not even 200 of them are found with the search terms of the present study. Combining the two would more than double the available publication set. In other words, both of the search formulations do only find a part of the relevant corpus, or one or both are not suitable at all. A reliable publication set for the topic in question must be the basis for all the analyses and visualizations done afterwards and would only validate the conclusions.
3. Results:
3.1. Fig.3 Is there really a difference between the growth rates of the second and the third period?
3.2. The authors obviously show in Fig 4. a citation analysis of sources provided by VOSviewer, but it is not presented as such and not used to draw any conclusions concerning the topic “resilience” and therefore not needed. The numbers in Table 2 seem to be derived from the associated map file, the significance of the “total link strength” is not explained, and the (Journal) Impact Factor is added without denoting its source and significance - which moreover is debatable.
3.3. Collaboration networks: The VOSviewer map Fig. 5 is only plotted without discussing its significance or actual collaborations. Do the values in table 3 represent indicators from the whole WoS or the publication subset of 1148 documents file? In the first version of the manuscript some of the authors have more publications. This is another indication that the search formulation must be defined more comprehensive and explained concerning its outcome in terms of recall and precision.
Fig 7, the VOSviewer map of institutional co-authorship, again is not discussed concerning the collaborations but only in terms of numbers of documents. And why pointing out China, having only one top-17 institution (line 192) or Tsinghua University with only 35 citations (line 196)? And how do the authors come to the conclusion in line 197? (Which by the way is not well placed in a results part.)
3.4. (Co-)citation analysis: The methodology is not explained, neither are the significance of most frequently cited papers over against those with highest total link strengt. The respective papers are named but not discussed in their significance
3.5. Keyword hotness analysis: Here, the different views on the co-occurence map in Fig 9 and 10, as well as the temporal density maps in Fig. 11 are helpful. But they could have been more focused if the VOSviewer parameter had been tuned to show only four or even three clusters which could be interpreted more comprehensively.
4. The “Conclusions” at first summarize some of the main results, but from line 317 on they contain some claims (“some positive aspects”?) which are not clearly derived from the results, especially “controversies” and “drawbacks” in theory. From line 325 onwards the authors present some predictions that are not reasonably derived from their results.
Author Response
June 2, 2022
Dear reviewer,
Thank you for your letter and the opportunity to revise our paper titled “Analysis and visualization of research on resilient cities and communities based on VOSviewer”. The comments and suggestions from you helped us enhance the quality of the work. The manuscript has been thoroughly revised in line with the comments. This report provides a detailed response to each comment and points out the revisions we have made. For a clear presentation, the comments are in red with italic font, our responses are in black, and the revisions in the revised manuscript are highlighted in red.
With kind regards,
The authors
Reviewer #1: The current revision is a very different manuscript, has been improved in some aspects, but still has significant deficits in methodology and presentation.
The main differences/improvements/criticism:
- The authors dropped their focus on Chinese research retrieved from the Chinese database CNKI which makes the investigations more reproducible. The title is changed accordingly. But it is still not as clear as it should be. I think the authors mean this: “Analysis and visualization of research on resilient cities and communities based on VOSviewer”.
Thank you for your kind comment. We improved the entire manuscript. And the title has been changed accordingly.
- Introduction:
a.) This part is significantly improved and now contains many more helpful explanations and references on the topic of resilience as well as on the bibliometric methods that the author are going to use and for what purpose. But on line 64 I suggest a different wording for clarity: “One important part of bibliometrics are citation visualization analysis methods gradually …”
Thank you for your suggestion. We modified the wording on line 64 for clarity in the revised manuscript. (Page 2 of the revised manuscript)
b.) And one important reference is missing: The review article “Resilient City: A Bibliometric Analysis and Visualization” by Yang et al (2021), https://doi.org/10.1155/2021/5558497. Yang et al. seem to have undertaken very much the same endeavour as the present authors and it is a “must” to interact with their approach, their methods (retrieval formulation, visualization using CiteSpace) and obviously different results(!). The authors must give clear reason why they even performed their investigations in the first place, when there already was such a review, and what they wanted to improve or which was their different scope. I’m sure the authors knew this work, because they nearly cite it on lines 127 to 129 and have the same division in time periods from 2010 on in Fig 3 and on lines 125 to 127. But they are not giving the reference which is no good scientific practice!
Thank you for your question and suggestion. The review article from Yang et al (2021) had been added to the revised manuscript. It was our neglect that failed to introduce into the reference. We already gave the reference.
In the section Introduction, we added the analysis and comparison with the existing review so that readers could better understand the work of this paper. (Page 2 of the revised manuscript)
- Materials and Methods:
c.) For the search procedure in WoS, the authors took my suggestion and described it in a bit more detail. But a screenshot of the WoS web interface has no additional information content! And my suggestion was by no means meant as a comprehensive search strategy in order to maximize recv all and precision of the topical search. This method of retrieval is still a main weakness. Again my own retrieval on 19 May 2022 (restricted to indexing dates until four weeks ago) got different results: more than 1,300 documents. So the authors applied some restriction, probably excluding the book indices, which they should mention and explain. Then Table 1, listing the different document types, is comparable to my results. Moreover, it is better practice to restrict the publication year to the last full year for bibliometric studies, in this case 2021. This makes the results more reproducible and presentable, cf. Fig 3.
Thank you for your question and suggestion.
Firstly, we deleted the screenshot of the WoS web interface. Secondly, We have a lot of talks about the selection of search strategy and considered that this current search formulation is a valuable way. We are very grateful for your valuable suggestions. Of course, this method of retrieval has still a few flaws, but we believed this will be an important reference for future researchers in this domain. The different and more precise search formulation is a direction worth exploring in future studies. Thirdly, Again my own retrieval on 31 May 2022 got 1167 documents according to the set search method. Because some articles published online were also searched.
What`s more, some restrictions were applied to exclude some irrelevant literature. We added this content in the revised manuscript.
Finally, We are very grateful for your valuable suggestions. “it is better practice to restrict the publication year to the last full year for bibliometric studies”. We will take note of this recommendation in future bibliometric studies. Consider not changing the date of the first search and originality of data, we did not change the publication year, but the analysis in this paper is essential to 2021. (Page 2-3 of the revised manuscript)
d.) Their research procedure is now clearer described in Figure 2 (in the first box I would not talk of “data mining”, it is simple “retrieval”).
Thank you for your suggestion. Figure 2 had been modified in the revised manuscript. (Page 3 of the revised manuscript)
e.) As mentioned above Yang et al (2021) have a different approach, in particular restricting their time span to publications years after 2010, because the former year have too few papers. Moreover, they use a different retrieval method using some keywords. Trying to reproduce their search and also excluding the book indices I got more than 1,500 papers between 2010 and 2019– not even 200 of them are found with the search terms of the present study. Combining the two would more than double the available publication set. In other words, both of the search formulations do only find a part of the relevant corpus, or one or both are not suitable at all. A reliable publication set for the topic in question must be the basis for all the analyses and visualizations done afterwards and would only validate the conclusions.
Thank you for your question and suggestion. We believed the different and mixed search formulation is a direction worth exploring in the subsequent work. Yang et al (2021) and this paper will be important references for future researchers in this field. The current search formulations are also meaningful.
- Results
3.1. Fig.3 Is there really a difference between the growth rates of the second and the third period?
Thank you for your question. There is a difference between the growth rates of the second and the third period. We roughly divided it into 3 stages. From 1995 to 2004, there were almost no relevant articles published in this field. After 2004, the number of research papers in this field showed a general trend of steady growth, but before 2014, the number of the published article presents a trend of stable increase. After 2014, With the acceleration of global urbanization and the increase of various natural and human disasters, the vulnerability of cities has become increasingly obvious. The period belongs to the rapid growth stage.
3.2. The authors obviously show in Fig 4. a citation analysis of sources provided by VOSviewer, but it is not presented as such and not used to draw any conclusions concerning the topic “resilience” and therefore not needed. The numbers in Table 2 seem to be derived from the associated map file, the significance of the “total link strength” is not explained, and the (Journal) Impact Factor is added without denoting its source and significance - which moreover is debatable.
Thank you for your question and suggestion. We deleted Fig 4. in the revised manuscript. In Table 2, we deleted the column of the “total link strength”. The (Journal) Impact Factor was a measure of a journal's impact capacity and retrieved from the 2021 Journal Citation Reports and added at the bottom of Table 2 in the revised manuscript. Although the impact factor is not a determinant of a journal, it is a valuable reference. Therefore, we added this content. (Page 5 of the revised manuscript)
3.3. Collaboration networks: The VOSviewer map Fig. 5 is only plotted without discussing its significance or actual collaborations. Do the values in table 3 represent indicators from the whole WoS or the publication subset of 1148 documents file? In the first version of the manuscript some of the authors have more publications. This is another indication that the search formulation must be defined more comprehensive and explained concerning its outcome in terms of recall and precision.
Thank you for your question and suggestion. We added and discussed the significance and actual collaborations of Fig.5. Table 3 represents indicators from the whole WoS. We thought a lot about the choice of search formulation and considered that this current search formulation is a valuable way, which is helpful for researchers. We are very grateful for your valuable comments. Surely, we believe the different and more precise search formulation is a direction worth exploring in future research. ( Page 6-8 of the revised manuscript)
Fig 7, the VOSviewer map of institutional co-authorship, again is not discussed concerning the collaborations but only in terms of numbers of documents. And why pointing out China, having only one top-17 institution (line 192) or Tsinghua University with only 35 citations (line 196)? And how do the authors come to the conclusion in line 197? (Which by the way is not well placed in a results part.)
Thank you for your question and suggestion. Firstly, we added and discussed the institutional co-authorship. At the same time, we deleted the description about pointing out China and rewrote the sentence. Additionally, the original conclusion in line 197 was updated according to the findings. ( Page 8 of the revised manuscript)
3.4. (Co-)citation analysis: The methodology is not explained, neither are the significance of most frequently cited papers over against those with highest total link strengt. The respective papers are named but not discussed in their significance
Thank you for your question. In the revised manuscript. we added the content and the significance of the most frequently cited papers over to those with the highest total link strength. We also discussed their significance instead of describing this map. (Page 9 of the revised manuscript)
3.5. Keyword hotness analysis: Here, the different views on the co-occurence map in Fig 9 and 10, as well as the temporal density maps in Fig. 11 are helpful. But they could have been more focused if the VOSviewer parameter had been tuned to show only four or even three clusters which could be interpreted more comprehensively.
Thank you for your question and suggestion. We re-drawn and interpreted the map which shows only three clusters in the revised manuscript. (Page 10-13 of the revised manuscript)
- Discussion
The “Conclusions” at first summarize some of the main results, but from line 317 on they contain some claims (“some positive aspects”?) which are not clearly derived from the results, especially “controversies” and “drawbacks” in theory. From line 325 onwards the authors present some predictions that are not reasonably derived from their results.
Thank you for your question. In the section “Conclusions and Discussions”, we added some discussion about RC research. And we deleted some controversial claims and some predictions that are not reasonably derived from their results in the revised manuscript. (Page 13-14 of the revised manuscript)
We would like to thank you again for taking the time to review our manuscript.

Reviewer 2 Report
I accept the author's answers to my comments and recommendations. In my opinion, the revised version of the manuscript has improved a lot, the inconsistencies and other problems have been fixed by the authors.
Author Response
June 2, 2022
Dear reviewer,
Thank you for your letter and the opportunity to revise our paper titled “Analysis and visualization of research on resilient cities and communities based on VOSviewer”. The comments and suggestions from you helped us enhance the quality of the work.
With kind regards,
The authors
Reviewer #2: I accept the author's answers to my comments and recommendations. In my opinion, the revised version of the manuscript has improved a lot, the inconsistencies and other problems have been fixed by the authors.
Thank you for your kind comment and acceptance. We would like to thank you again for taking the time to review our manuscript.
Round 2
Reviewer 1 Report
The authors took up most of my suggestions and points of criticism and changed resp. expanded their manuscript accordingly.
There are only some minor issues remaining:
The new paragraph starting in line 78 and dealing with Yang (2021) should be edited with respect to English language use.
3.3. Collaboration networks: The discussion would be more understandable if the authors would provide online versions of VOSviewer maps Fig 3 and 4 because the names they list are not all findable in the printed version.
Line 221: The caption of Fig. 5 is unclear. And the conclusion in line 215 drawn from Fig.5 should be clarified.
Author Response
June 6, 2022
Dear reviewer,
Thank you for your letter and the opportunity to revise our paper titled “Analysis and visualization of research on resilient cities and communities based on VOSviewer”. The comments and suggestions from you helped us enhance the quality of the work. The manuscript has been thoroughly revised in line with the comments. This report provides a detailed response to each comment and points out the revisions we have made. For a clear presentation, the comments are in red with italic font, our responses are in black, and the revisions in the revised manuscript are highlighted in red.
With kind regards,
The authors
Reviewer #1:(Round 2) The authors took up most of my suggestions and points of criticism and changed resp. expanded their manuscript accordingly.
Thank you for your kind comment. We improved the entire manuscript.
There are only some minor issues remaining:
The new paragraph starting in line 78 and dealing with Yang (2021) should be edited with respect to English language use.
Thank you for your suggestion. We modified and edited the new paragraph starting in line 78 in the revised manuscript. (Page 2 of the revised manuscript)
3.3. Collaboration networks: The discussion would be more understandable if the authors would provide online versions of VOSviewer maps Fig 3 and 4 because the names they list are not all findable in the printed version.
Thank you for your suggestion. We updated and provided online versions of VOSviewer maps Fig 3 and Fig.4 in the revised manuscript. (Page 6-8 of the revised manuscript)
Line 221: The caption of Fig. 5 is unclear. And the conclusion in line 215 drawn from Fig.5 should be clarified.
Thank you for your suggestion. We restated Fig.5. In the revised manuscript, we deleted the conclusion in line 215. (Page 8 of the revised manuscript)

This manuscript is a resubmission of an earlier submission. The following is a list of the peer review reports and author responses from that submission.
Round 1
Reviewer 1 Report
The authors aim at presenting a bibliometric study of the research on resilient cities and communities that has been published between 2001 and 2020 and indexed in two different bibliographic databases – the CNKI containing Chinese publications and the WoS. As main tool for the visualization of the results, they used the VOSviewer which they mentioned prominently in the title. (This title is by the way not clearly formulated.)
In my view the authors lack an understanding of bibliometrics and the use and significance of its tools and fail to present a thorough bibliometric investigation of the research field in focus.
In the following I go through the manuscript to present my criticism in more detail, indicating the section and the paragraph by “P.” because the manuscript unfortunately had no line numbers.
1. Introduction:
- P.1: I cannot infer from Ref. [1] the concept of resilience upcoming in the 1970s.
- P.2: COVID-19 no example for the three threats form first sentence;
- P.2: The authors cite Ref. [7] from 2018 to talk about “long”-term concern ...
- P.3: The discipline of bibliometrics is misrepresented: it is not only the use of visualization tools as VOSviewer or CiteSpace – which is cited in Ref. [13] - but contains a lot of reasoning and (statistical) methods. The actual methods used in the Refs. [13] to [18] should be mentioned and discussed. What do the authors mean by the “traditional literature review method” and is ist applied in these Refs.?
- P.3: What are “research laws”?
2. Materials and Methods:
-P.1: How did the authors get access to CNKI? A description of available metadata is lacking. The search procedure is not described in detail. Why RefWorks as output format?
- P.1: I very much doubt the validity of the search in WoS ( and would infer the same problems with CNKI, too): Firstly, the search term are misspelled: resilient - not resident! Secondly, they should be truncated in order to retrieve plurals, too. But even with that improvement, my own search on March 21, 2022, with TS= ("resilient cit*" or "resilient communit*") only yielded 1,282 Documents ( without truncation only 421), not excluding any document type. And why should one exclude conference papers? So I doubt the data the authors use for their study.
-P.2: Again confusion of bibliometrics with tools like VOSviewer. And why does it perform an “econometric analysis”?
- Figure 1: in the left upper box: there is no such thing as “bibliometric search” but document retrieval.
How is the de-duplication done? Are English synomyms removed in order to get a pure Chinese dataset? How can VOSviewer after the restriction on a number of keyword occurrences also restrict the number of author documents? This is probably an alternative evaluation and no further step in the flow chart.
Please use “piece of literature” or paper but not “literature” alone., in the whole manuscript.
3.Results
3.1.
- P.2.: Concerning Fig.2, what are the numbers until 2011?
“In the stage of 2012 to 2014, the cumulative number of articles published is 5,” so no growth at all??
3.2.1.
Fig 4. seems to me to be over-interpreted. In order to name Lu Xinzheng as most important (“central”) author one should investigate his role in the 3(!) publications he co-authored. And Zahi Guofang with the most (6) publications is not to be found in the map.
3.2.2.
Fig. 5, the corresponding VOSviewer map, is also over-interpreted. The contrast to Fig.4 can mainly be an artefact of the very different numbers of publications and authors. The number of citations is given in Table 2, but not in Table 1. Here and in other places the authors only drop numbers and do not discuss their significance for the research field.
3.3.1
At the end of this section the authors mix results with recommendations.
3.3.2.
P.1.: “number of articles” should read “number of citations”
Last sentence: Does a VOSviewer map like Fig.7 warrant such conclusions? Can the authors cite references that give reasons for that? By the way, the VOSviewer maps should be provided as online versions for further inspection.
3.4.1
That the most cited documents are often reviews and similar document types is a well-known fact. The papers are only listed and not discussed in their relevance.
3.4.2.
How were 175 documents chosen out of 949?
Fig. 8 is a network of documents, not research institutions. It is not described in the text, that the size of the nodes is correlated with citation numbers. The highly-cites reviews are only listed, even with their linkage strengths, but the relevance of these properties is not discussed. Especially the relevance of Allison [25] would be interesting.
3.5.1
(4) I do not see “Climate change” in the first place! What is “research fervor”?
3.5.2.
Fig.10 : the shape of the co-occurrence network as opposed to Fig. 9 can again be an artefact of the very different numbers of publications and therefore keywords, as well.
That Resilience is the core keyword is clear from the initial search for this term!
4. Discussion
This is no Discussion, but a list of recommendations that the authors draw from some of their results.
5. Conclusions
The “Conclusions” are only a summarizing list of results.
(4) Are the two bibliographic databases really comparable, especially in the coverage of citations?
And why do the authors draw a connection between number of citations and depth of research – whatever that should be?
Reviewer 2 Report
Introduction
- the presentation of the concept of resilience is rather rough, so it does not turn out that its meaning covers different things in each research area, so when we talk about resilient cities, it can be interpreted in many ways, so a deeper introduction to the concept is necessary, please see the following literatures:
- Forrester, J. W. (1969): Urban Dynamics. Cambridge, Massachusetts, Massachusetts Institute of Technology Press, 285 p.
- Walker, B., Holling, C. S., Carpenter, S., Kinzig, A. (2004): Resilience, adaptability and transformability in social–ecological systems. Ecology and society, 9(2, article 5), DOI: 10.5751/ES-00650-090205
- Folke, C. (2006): Resilience: The emergence of a perspective for social–ecological systems analyses. Global environmental change, 16(3), 253-267.
- Meerow, S., Stults, M. (2016). Comparing conceptualizations of urban climate resilience in theory and practice. Sustainability, 8(7), 701. DOI: 10.3390/su8070701
- Meerow, S., Newell, J. P., Stults, M. (2016). Defining urban resilience: A review. Landscape and urban planning, 147, 38-49. DOI: 10.1016/j.landurbplan.2015.11.011
- more defined research aims are needed, there are no real research questions, it seems the authors have randomly chosen this area as the subject of their bibliometric analysis
Materials and methods
- the search rule is wrongly defined resident city or resident community instead of resilient city and resilient community,
- -the authors do not describe what the keywords verification means (shows on Fig. 1.), how it was performed, how the keywords were improved,
- the citation analysis part of the research is missing from the methods and materials section (for example from Fig. 1.), how these articles were selected, in the Chinese literature the top ten cited articles were chosen but in the case of foreign analysis a minimum citation count was set, why the difference in the selection method (I don’t say this isn’t good, but the reader doesn’t have information about the considerations), but the main problem with this part that it can be misunderstood because the reader can also think about the analysis of the most cited references in this context
Results
- the first paragraph belongs to methods and is repetitive in nature,
- in the abstract, the authors claim that the development of literature about resilient cities and resilient communities has two stages whereas, in the results section, they identified three stages (at least in Chinese literature): between 2001-2011, between 2012-2014, and one after 2014, so which one is correct?
- in 3.2.1. & 3.2.2 subsections, the authors describe the figures and tables in textual descriptive form, but a more substantive analysis is needed here about these cooperation networks (for example the research fields of these, the timeframe when they were active and so on)
- subsection 3.3 is also too descriptive
- in section 3.4.1 the 2nd sentence belongs to the methods section
- section 3.5 is interesting and the most valuable part of the article so far
Discussion and Conclusion sections
- because of the short discussion and because certain parts of the conclusion are also part of the discussion, I suggest to the authors combine the two sections
(Please use row numbering next time, it is much easier to add precise comments to the text.)
Otherwise, the structure of the article is logical, and the methods and software used are appropriate for the scientometric analysis. The article is well illustrated and easy to read.
Based on the above, I suggest a major revision of the manuscript.
